# Personalized, Precision Medicine to Cure Alzheimer’s Dementia: Approach #1

**DOI:** 10.3390/ijms25073909

**Published:** 2024-03-31

**Authors:** Jeffrey Fessel

**Affiliations:** Clinical Medicine, Department of Medicine, University of California, 2069 Filbert Street, San Francisco, CA 94123, USA; w.fessel@ucsf.edu; Tel.: +1-415-563-0818

**Keywords:** Alzheimer’s dementia, cure of dementia, 18 elements of pathogenesis, drugs addressing each element, sequential administration of triple-drug treatments

## Abstract

The goal of the treatment for Alzheimer’s dementia (AD) is the cure of dementia. A literature review revealed 18 major elements causing AD and 29 separate medications that address them. For any individual with AD, one is unlikely to discern which major causal elements produced dementia. Thus, for personalized, precision medicine, all causal elements must be treated so that each individual patient will have her or his causal elements addressed. Twenty-nine drugs cannot concomitantly be administered, so triple combinations of drugs taken from that list are suggested, and each triple combination can be administered sequentially, in any order. Ten combinations given over 13 weeks require 2.5 years, or if given over 26 weeks, they require 5.0 years. Such sequential treatment addresses all 18 elements and should cure dementia. In addition, any comorbid risk factors for AD whose first presence or worsening was within ±1 year of when AD first appeared should receive appropriate, standard treatment together with the sequential combinations. The article outlines a randomized clinical trial that is necessary to assess the safety and efficacy of the proposed treatments; it includes a triple-drug Rx for equipoise. Clinical trials should have durations of both 2.5 and 5.0 years unless the data safety monitoring board (DSMB) determines earlier success or futility since it is uncertain whether three or six months of treatment will be curative in humans, although studies in animals suggest that the briefer duration of treatment might be effective and restore defective neural tracts.

## 1. Introduction and Background

Despite thousands of articles that detail its causal factors, there is still no cure for Alzheimer’s dementia (AD). A likely reason for this failure is that almost all clinical trials of potential treatments have applied only single drugs. There are, however, two generic principles applicable to curing any medical condition. For AD, the first of those principles would treat all of the known possible causes that contribute to its pathogenesis. AD has a large number of causal factors; the precise number depends upon how those causes are counted. That number, however counted, is large, and potentially curative treatment requires, correspondingly, a very large number of drugs. How to apply those drugs is a major consideration, but in this first approach, the same potential cure would apply to every patient. Because not all potential causes affect every patient, the second generic principle identifies for each separate patient those components of pathogenesis that are actually operative and treats only those; this creates personalized, precision medicine. Each approach has merit. This article describes the first approach, whose underlying concept is that an assured cure for dementia requires counteracting all of the known pathogenetic components, some of which inevitably affect each individual patient. By using treatments that address all potentially causal elements, each individual patient will assuredly have her or his causal elements addressed, which also creates personalized, precision medicine.

## 2. Methods of Approach and Results

Literature searches using PubMed and Google Scholar, plus citation lists of appropriate articles, revealed 18 major elements that contribute to the pathogenesis of AD (see Table 1) and 29 drugs needed to counteract them. Those major elements include depositions of (1) Aβ and (2) tau proteins, (3) oxidative stress and reactive oxygen species (ROS), (4) reduced level of brain-derived neurotrophic factor (BDNF), (5) decreases in the levels of TGF-β and (6) Wnt/β-catenin, (7) alteration of the epithelial-mesenchymal transition (EMT), (8) cerebral microcirculatory abnormalities, (9) insulin resistance and reduced brain glucose utilization, (10) neuroinflammation, (11) excessive intracellular calcium, (12) genetic effects, particularly of *APOE4ε*, and epigenetic effects, (13) impaired unfolded protein response (UPR), (14) reduced autophagy, (15) changes induced by microRNAs (miRNA), (16) mitochondrial dysfunction, (17) abnormal circadian rhythmicity, (18) and decreases in the number and functions of neurons and synapses, affecting the neural networks whose impairment is a feature of AD. The following shows the drugs required to address each of the above elements. Although the drugs cannot all be administered simultaneously, their sequential administration over 2.5 or 5.0 years, in combinations of three drugs, permits addressing all applicable components of pathogenesis, thus assuring that each patient receives the drugs that reverse the major causes of her or his dementia. That is why this approach is personalized and precise.

## 3. The 29 Treatments for the 18 Components of Pathogenesis

### 3.1. Treatment to Reverse the Deposition of Aβ

Anti-Aβ treatments include lecanemab, aducanumab, and donanemab; although they failed to cure AD, the use of partner drugs that benefit components besides amyloid might potentially induce a cure. The partner drugs must address the unfolded protein response (UPR), autophagy, mitophagy, reactive oxygen species (ROS), the Wnt/β-catenin pathway, hyperphosphorylated tau protein, and impaired calcium homeostasis; the drugs effective for those are listed in the following sections.

*Drugs useful for reversing the deposition of Aβ* include anti-amyloid compounds, e.g., lecanemab, aducanumab, or donanemab.

### 3.2. Treatment to Reverse the Deposition of Tau Protein

Aggregates of hyperphosphorylated tau protein form the paired helical filaments in neurofibrillary tangles; they impair neural function by interfering with microtubules that transport metabolites down axons. Rapamycin inhibits the hyperphosphorylation of tau [1]. Phosphorylation is also caused by GSK3β, which is prevented by lithium and valproate [2]. Lecanemab produces a decrease in pTau181 [3]; donanemab, which may soon become commercially available, produces a decrease in the levels of pTau217 [4]. Lithium also upregulates Bcl-2, which is cytoprotective and antiapoptotic [5]. Both rapamycin and nicotinamide also inhibit tau hyperphosphorylation [1,6].

*Drugs useful for reversing the deposition of tau protein* are rapamycin, lithium, valproate, lecanemab, donanemab, and nicotinamide.

### 3.3. Treatment to Reverse Reactive Oxidative Stress (ROS)

ROS become a risk factor for AD when the antioxidant system is inadequate. ROS include superoxide anion radical, hydrogen peroxide, hydroxyl radical, singlet oxygen, alkoxyl radicals, peroxyl radicals, and peroxynitrites. ROS occur in AD for numerous reasons: Aβ may bind metal ions such as copper when, via an electron transfer mechanism, hydroxyl radicals form; hyperphosphorylated tau and associated inflammatory responses induce ROS, and so do lipid peroxidation, protein oxidation, and nucleic acid oxidation, which cause mitochondrial dysfunction with the production of ROS. Several pathways activated by ROS produce neuronal apoptosis and AD.

Nicotinamide reduces levels of ROS [7]; antioxidant vitamins E and C give a lower incidence of AD in the elderly population [8]; and vitamin E slows the progression of moderately severe established AD [9]. Vitamin A is an antioxidant, but potential adverse effects limit its use except in low dosages.

*Drugs that address ROS* are nicotinamide and vitamins E and C.

### 3.4. Treatment to Reverse the Reduced Level of Brain-Derived Neurotrophic Factor (BDNF)

BDNF could repair the ~70% depletion in CA1 hippocampal neurons that is seen in AD [10]. Its poor BBB penetration is overcome by its conjugation with clathrin [11], with a 400-fold increase in hippocampal BDNF [12]; however, that conjugate is commercially unavailable. Several reports show that pioglitazone increases BDNF [13,14,15]. One study showed that pioglitazone restored BDNF levels by 83% of normal after exposure to a PPARγ antagonist [15]. Peroxisome proliferator-activated receptor gamma (PPARγ) agonists such as pioglitazone provide several other benefits: they prevent the phosphorylation of JAK–STAT in astrocytes and, thereby, induce increases in neurons and endothelial cells; they upregulate M2 microglia, whose ratio relative to proinflammatory M1 microglia thus increases; they also reverse the decreased expression of PPARγ receptor in astrocytes after exposure to Aβ25–35. PPARγ agonism increases the neurogenic differentiation gene NeuroD1 and protects cortical neurons and axons against toxicity induced by nitric oxide (NO) or potassium chloride (KCl) [16,17].

A drug useful for reversing the reduced level of BDNF is pioglitazone.

### 3.5. Treatment to Reverse Reduced Levels of Transforming Growth Factor β (TGFβ)

TGF-β levels in AD are inadequate, and several available drugs raise them (see ref. [18]). Its levels increase after the administration of selective serotonin reuptake inhibitors (SSRIs) for depression; adding fluoxetine to cultures of glia and neurons prevented Aβ_1–42_-induced neural toxicity but did not do so in cultures of only neurons [19]. Because astrocytes produce TGF-β, they are exposed to fluoxetine, which cause TGF-β secretion, and the culture medium protected neurons from Aβ-induced toxicity. Confirming the role of astrocytes and the benefit of SSRI drugs, venlafaxine caused no increase in neuronal levels of TGF-β but did induce a 35% increase in astrocytic levels in the penumbra of cerebral infarction [20]. Memantine also induces an increased production of TGF-β. In a randomized controlled trial (RCT), patients taking methadone used either low doses of memantine or placebo [21]. After 12 weeks, those taking memantine had higher levels of TGF-β and required lower doses of methadone. Finally, intranasal insulin has direct access to the brain, and after its administration, TGF-β receptors translocated from intracellular stores to the plasma membrane, thus beneficially for AD, enhancing responsiveness to TGF-β [22].

*Drugs useful for reversing the reduced levels of TGF-β* include fluoxetine, memantine, and intranasal insulin.

### 3.6. Treatment to Reverse the Reduced Levels of Wnt/β-Catenin

The pro-synaptogenic effects of Wnt proteins result from their activation of the canonical Wnt signaling pathway, whereas the non-canonical pathway is anti-synaptogenic [23] since the canonical pathway involves β-catenin whereas the non-canonical pathway does not. When activated, β-catenin translocates to the nucleus and regulates the transcription of beneficial genes. Activated GSKβ promotes the degradation of β-catenin; lithium inhibits GSKβ activation and is a Wnt/β-catenin agonist of the canonical pathway [24]. Since AD has reduced levels of Wnt/β-catenin [25], lithium’s administration would be beneficial. The upregulation of Wnt/β-catenin via the canonical pathway is achieved with doxycycline [26,27], simvastatin [28], and glucocorticoids [29].

*Drugs useful for reversing the reduced level of Wnt/β-catenin* include lithium, doxycycline, simvastatin, and glucocorticoids.

### 3.7. Treatment to Reverse the Direction of EMT from M-E to E-M

EMT is the process by which cells change their phenotype from one typifying an epithelial lineage (E) to one typifying a mesenchymal lineage (M). The transition may be either E-to-M (EMT) or M-to-E (MET). Cadherins are involved in this and label the neuronal phenotype as either E or M [30]. Most brain neurons have the E phenotype and express E-cadherin or have the M phenotype and express N-cadherin.

In AD, M phenotype neurons are a minority, and E neurons dominate. FAM3C is a key molecule in causing the E-to-M transition; it was 45% lower (i.e., fewer M phenotypic neurons) in AD brains than in controls [31] and was associated with the Braak stage: −27% in Braak stages 3–4 but −51% in Braak stages 5–6 [31,32]. TGF-β induced the neuronal expression of FAM3C [33]; thus, raising levels of TGF-β will also increase EMT.

Wnt/β-catenin itself enhances EMT: adding it to epithelial cell lines stabilized the complex between cytoplasmic β-catenin and lymphoid enhancing factor-1 (LEF-1), which then transports it to the nucleus, leading to EMT; overexpressed LEF-1 dramatically promotes EMT [34]. Thus, treatments raising Wnt/β-catenin will also raise EMT.

*Drugs useful for reversing the direction of the E-M transition* include those that increase levels of TGF-β, which are fluoxetine, memantine, and intranasal insulin, and of Wnt/β-catenin, which are doxycycline, lithium, simvastatin, and corticosteroids.

### 3.8. Treatment to Reverse the Cerebral Microcirculatory Abnormalities

The disrupted blood–brain barrier (BBB) participates in the pathogenesis of AD. Valproate induces clusterin, which blocks the transport of Aβ across the BBB [35], and valproate facilitates endothelial cell proliferation, thus increasing CBF [36]. Further, valproate elevates histone acetylation by inhibiting histone deacetylase and, by affecting chromatin remodeling, histone hypoacetylation contributes to cognitive impairment in AD. Therefore, the deinhibition of histone acetylation induced by histone deacetylases (HDAC) inhibitors contributes to the recovery of learning and memory. That is the probable mechanism by which valproate enhanced long-term recognition memory and spatial learning and memory in AD transgenic mice [37].

A drug useful for reversing cerebral microcirculatory abnormalities is valproic acid.

### 3.9. Treatment to Reverse Insulin Resistance in the Brain

Because type 2 diabetes is a risk factor for AD, the role of insulin has been extensively studied. Patients with MCI who progressed to dementia had a reduced cerebral metabolic rate of glucose metabolism [38]; that is because cerebral resistance to insulin is heightened in diabetics [39]. An additional mechanism of insulin’s action is the reduction of the proinflammatory M1 microglial phenotype [40].

### 3.10. Treatment to Reverse Neuroinflammation

Amyloid deposits activate microglia, which then release neurotoxins; amyloid also stimulates an overall immune response [41]. For AD treatment, many nonsteroidal anti-inflammatory agents (NSAIDs) have been used with highly variable responses [41,42]. Diclofenac is the only NSAID associated with a significant reduction in cognition. A report of a retrospective cohort study of 1431 patients with AD receiving diclofenac showed a reduced incidence of dementia as compared to other NSAIDs; that report discussed the mechanisms whereby diclofenac is effective [42].

A drug useful to reverse cerebral inflammation is diclofenac (extended-release) 100 mg qd.

### 3.11. Treatment to Reverse Increased Intracellular Ca^2+^ Levels

Enhanced intracellular Ca^2+^ levels may cause neurodegeneration in AD [43]. Notably, many of the genes that heighten susceptibility to AD alter intracellular calcium signaling [44], which contributes to neuronal degeneration and apoptosis, as well as to impaired presynaptic neurotransmitter release, postsynaptic signaling, and long-term synaptic plasticity.

Two different Ca^2+^-releasing mechanisms allow for the emptying of the main intracellular Ca^2+^ stored in the endoplasmic reticulum (ER). The first mechanism involves ryanodine receptors (RyR) in the ER, which are activated by increases of cytosolic Ca^2+^. The AD brain has elevated levels of intracellular Ca^2+^ partly because the plasma membrane Ca^2+^-ATPase and Na^+^/Ca^2+^ exchanger are reduced [45]. RyR is blocked either by dantrolene or when calmodulin binds to RyR2 [46]. The second mechanism for the release of Ca^2+^ involves inositol (1,4,5)-triphosphate receptors (IP_3_R) in the ER. In AD models, IP_3_R is enhanced, so its inhibition may help to prevent neuronal loss [47]. The two mechanisms for the release of Ca^2+^ are linked because RyR may induce the activation of IP_3_R.

B-cell lymphoma 2 (Bcl-2) family proteins play a crucial role in targeting intracellular Ca^2+^-transport systems and channels that mediate Ca^2+^-flux from the ER into the mitochondria [48]. IP_3_R is inhibited when bound to Bcl-2, which prevents the Ca^2+^ signaling that mediates cell death. Another protein, Mcl-1, has sequence similarity to Bcl-2 and also inhibits IP_3_R. Beyond their role as modulators of ER-resident Ca^2+^ channels, Bcl-2 and Mcl-1 interact with the voltage channel, VDAC1, to inhibit the permeabilization of the outer mitochondrial membrane induced by Ca^2+^ overload and thus prevent apoptosis caused by mitochondria, producing cytochrome C and caspase-9, which lead to apoptosis [49].

Treatments to address the heightened level of intracellular Ca^2+^ might include dantrolene, heparin, rasagiline, and caffeine. Dantrolene has caused fatal hepatitis in elderly subjects, so it should not be used; moreover, heparin antagonizes IP_3_R, but its anticoagulant effect interdicts its use. Two further drugs that antagonize IP_3_R are rasagiline and caffeine. 

Rasagiline is an approved drug that is an agonist of Bcl-2 [50], so it would be an effective inhibitor of IP_3_R. Its dosage is 0.5–1.0 mg daily. Caffeine is a simple but effective treatment for inhibiting IP_3_R [51,52]. The mean amount of caffeine in six brands of tea, with a steep time of 5 min, was 42 mg [53]. A total of 175 mg/kg of caffeine, injected into mice, inhibited the actions of IP_3_R [52]. Since the mean weight of an average house mouse is 32 G, it received 175/1000 × 32 mg = 5.6 mg caffeine. Delivering an effective dose of caffeine requires 42/5.6 = 7.5 cups of tea; that would be approximately achieved every week by drinking one cup of tea daily.

Drugs to reverse the increased intracellular Ca^2+^ in AD include rasagiline and one glass of tea daily.

### 3.12. Treatment to Reverse Increased APOE4ε

Many genes and their associated mRNA and proteins are risk factors for late-onset AD; among the many relevant genes shown by a multi-center study, *APOE4* had huge significance (*p* = 3.3 × 10^−96^) [54]. Its mechanism involves Aβ aggregation and clearance, tau phosphorylation and aggregation, lipid metabolism, inflammation, altered neuronal repair, and synaptic plasticity [55].

Treatment for reducing the effect of *APOE4ε* could include rapamycin, which maintained brain function in carriers of *APOE4ε* [56], and intranasal insulin, which also benefitted *APOE4ε* carriers. The administration of short-acting intranasal insulin improved verbal memory in APOEε4 non-carriers but not in *APOEε4 carriers*; however, presumably because of its longer half-life, long-acting intranasal insulin given to AD patients produced memory improvement in *APOEε4* carriers but worsening in *APOEε4* non-carriers [57].

*Drugs useful to reverse increased APOE4ε* are rapamycin and long-acting intranasal insulin.

### 3.13. Treatment to Reverse Epigenetic Changes

Epigenetic mechanisms cause genes to turn on or off; they act via DNA methylation, histone acetylation, and non-coding RNAs. In a highly simplified explanation, DNA methylation controls gene expression and occurs in CpG sites; there, a cytosine residue is followed by a guanine residue, and a methyl group is added to cytosine, catalyzed by DNA methyltransferase, which transfers a methyl group from S-adenosyl-methionine (SAM), to form 5-methylcytosine that recruits DNA proteins. After the transfer of a methyl group, SAM is converted to S-adenosylhomocysteine (SAH), then SAH is hydrolyzed to homocysteine. The acceleration of the entire process creates hyperhomocysteinemia, which is present in AD [58,59]. Hyperhomocysteinemia is corrected both by a combination of folic acid and pyridoxine [60] and via nicotinamide phosphoribosyltransferase nicotinamide [61]; correcting hyperhomocysteinemia would contribute to the cure of AD.

*Drugs useful to reverse epigenetic changes* are folic acid 5 mg daily, pyridoxine 50 mg daily, and nicotinamide 1000 mg daily.

### 3.14. Treatment to Reverse Reduced Autophagy

Autophagy by autophagosomes to remove abnormal protein aggregates is reduced in early AD [62]. Insulin resistance occurs in early AD and promotes the formation of autophagosomes, in which Amyloid Processor Protein (APP) processing and Aβ generation occur. The mammalian target of the rapamycin (mTOR) pathway negatively regulates autophagy. By inhibiting mTOR, rapamycin increases autophagy and decreases Aβ pathology [63]. Nicotinamide also enhances autophagy [61].

*Drugs useful for reversing reduced autophagy include* rapamycin and nicotinamide, which enhance autophagy [61,64].

### 3.15. Treatment to Address the miRNAs Involved in AD

MicroRNAs (miRNAs) are small, non-coding RNAs that mediate post-transcriptional gene silencing by binding to complementary sequences in the target mRNAs [65]. Many miRNAs are deregulated in AD blood [66]; ten of those occurred at Braak stage III and were functionally associated with the immune system, cell cycle, gene expression, cellular response to stress, neuron growth factor signaling, Wnt signaling, cellular senescence, and Rho GTPase; however, because miRNA molecules contain only a few amino acids, a single miRNA may potentially interact with the DNA of multiple genes, so there could be many more associations.

Suppressive oligonucleotides change the effect of miRNAs; their transfection into glioma cells containing upregulated miR-21 led to caspase activation and apoptotic cell death [67]. This potential treatment is commercially unavailable. Otherwise, a useful drug is gemfibrozil, an agonist of Peroxisome Proliferator-Activated Receptor α (PPARα), which increased miR-107 levels, which are decreased in AD [68].

A drug useful for addressing the miRNAs involved in AD is gemfibrozil.

### 3.16. Treatments to Reduce the Impaired Mitochondrial Function in AD

In AD, the mitochondrial number is reduced [69], in morphologically abnormal [70,71], has fewer genes encoding subunits of the electron transport chain [72], and has decreased activities of the tricarboxylic acid (TCA) cycle [73]. The PPARγ coactivator 1-α that regulates mitochondrial biogenesis had reduced levels in AD [70].

Via nicotinamide adenylyltransferase and nicotinamide adenyldinucleotide (NAD) synthase, nicotinamide yields the dinucleotides NAD^+^ and NAAD^+^. In mitochondria, nicotinamide participates in energy metabolism by utilizing NAD^+^ in the electron transport chain for the production of ATP [74]. ATP synthase was reduced in Braak stages 1–11, i.e., in the earliest stages of AD [75]. Since ATP corrects impaired folding of proteins, its deficiency in AD enhances the presence of amyloid; however, nicotinamide increases mitochondrial ATP, so it decreases the amount of amyloid in AD.

Treatment with nicotinamide benefitted mitochondrial biogenesis [76] and mitochondrial function [77].

A drug for reversing mitochondrial dysfunction is nicotinamide.

### 3.17. Treatment to Address the Disturbed Circadian Rhythmicity in AD

An impaired sleep/wake cycle contributes importantly to causing AD. Weldemichael and Grossberg noted that in AD, nocturnal sleep disturbance is associated with the degree of dementia [78]. Impaired sleep/wake cycle is also seen in both persons at risk of future AD and in early AD.

In cognitively normal participants with a mean age of 62.9 years and a parental history of sporadic AD, and thus at risk for future AD, worse subjective sleep quality and daytime somnolence were associated in CSF with lower Aβ_42_ and Aβ_40_ and higher total tau:Aβ_42_ and *p*-tau:Aβ_42_ [79]. Others found that the progression of AD was mirrored by sleep/wake variables [80]. The mechanism linking decreased sleeping time, and AD is the reduced removal of potential toxins, particularly Aβ, because the awake brain has a reduced volume of the interstitial space. The cortical interstitial space was 66% higher in sleeping mice than in awake mice [81]. Another study showed hippocampal Aβ levels during sleep deprivation rising by 33.7% and, with sleep, immediately falling [82]. That diurnal fluctuation of Aβ in the brain’s interstitial fluid was also seen in humans: one night of unrestricted sleep led to a 6% decrease in Aβ_42_ levels, whereas sleep deprivation counteracted that [83], but in sleep-deprived, cognitively normal subjects, the mean levels of overnight CSF Aβ_38_, Aβ_40_, and Aβ_42_ increased above baseline levels by 30% [84]. Similarly to tau protein, in healthy men, plasma total tau increased by 17.5% in response to sleep loss as compared to only 1.8% during normal sleep [85].

In view of the above explanation, it is surprising that therapy with light is beneficial. Photobiomodulation (PBM) is the mechanism by which nonionizing optical radiation in the visible and near-infrared spectral range is absorbed by endogenous chromophores to elicit photophysical and photochemical events; moreover, PBM therapy (PBMT) is a photon therapy that uses nonionizing forms of light to cause physiological changes and therapeutic benefits [86]. Among 18 randomized controlled trials in persons with dementia, light therapy reduced night-time awakenings and enhanced sleep quality [87]. Infra-red laser (780–1100 nm) triggered increased cerebral blood flow; mitochondrial activation, which enhanced neuroprotection; activated N-methyl-D-aspartate receptor (NMDAR), which decreased intracellular overload of Ca^2+^ and prevented excitatory neurotoxicity; and reduced excitatory neurotransmission in the hippocampus [87]. In response to light at 808 nM, microglia exposed to Aβ switched from glycolysis to enhanced anti-inflammatory activity, and when microglia were co-cultured with neurons and exposed first to Aβ and then to light, ROS production was decreased with less neurotoxicity [88]. Light induces beneficial pathways, including extracellular signal-regulated kinase (ERK), mitogen-activated protein kinase (MAPK), and protein kinase B (Akt), and activates neurotrophic factors and secretases [89,90]. Exposing the frontal skull of rats to a light-emitting diode after the inoculation of the hippocampus with Aβ led to improvements in spatial memory and behavioral and motor skills [90].

Briefly, the apparently paradoxical effect of opposing therapeutic approaches occurs because each approach induces different mechanisms.

Drugs useful to address the disturbed sleep/wake cycle involved in AD include melatonin to induce sleep and light treatment.

### 3.18. Treatment to Reverse Reduced Synaptic and Neuronal Function

It is the disturbed network of neural connections that results in AD and must be addressed by any program aiming to cure dementia. To some extent, although incompletely, it involves the Default Modal Network (DMN), which, anatomically, comprises the medial prefrontal cortex (mPFC), the posterior cingulate cortex (pPCC), the precuneus (PC), and the angular gyrus (AG). Additional connections that are disrupted include the limbic network between the anterior temporal lobe and the orbitofrontal cortex, the ventral attention network, and the dorsal attention network [91]. Although a decrease in cholinergic neurons in the nucleus basalis of Meynert may be a primary event [92], by the time AD has developed, there are also decreases in both glutamatergic and gabaergic neurons. In AD, amyloid pathology progresses where cholinergic terminals appear most vulnerable, followed by glutamatergic terminals and finally by gabaergic terminals [93]. Since AD-model mice are frequently used as surrogates for AD in humans, it is worth mentioning that glutamate and GABA neurotransmission dominate in the mouse brain. Yao et al. reported neurotransmission in 31 regions of the mouse brain: it was gabaergic in 15, it was glutamatergic in 13, and it was cholinergic in only three, but such distributions are unavailable for the human brain [94]. In human AD, there are also pathways utilizing both adrenergic and serotonergic neurons. Drugs that are available to increase cholinergic, glutamatergic, gabaergic, adrenergic, and serotonergic neurons are mentioned in the following paragraphs.

(a)*Reduced activity of acetylcholine.* This can be treated by three FDA-approved anticholinesterase drugs, donepezil, galantamine, and rivastigmine, and also by pioglitazone, which restored anticholinesterase esterase to 83% from its reduction by an anticholinesterase agent [15]. The effect of anticholinesterase drugs on neural circuitry was demonstrated for both galantamine and donepezil. Galantamine increased functional connectivity in the posterior subcomponent of the DMN involving the PCC and PC [95], and donepezil increased connectivity in the parahippocampal gyrus and stabilized activation in the precuneus [96].(b)*Reduced glutamatergic function.* In AD, the activation of specific glutamate receptors may lead to neurotoxicity because extrasynaptic *N*-methyl-D-aspartate receptor (NMDAR) signaling, activated by glutamate released from astrocytes or presynaptic terminals, antagonizes synaptic pro-survival signaling and tilts the balance toward excitotoxicity and neurodegeneration [97]. NMDAR antagonists include memantine and riluzole. Memantine plus anticholinesterase drugs created a greater reduction in behavior disturbance than anticholinesterase drugs given alone [98]. Riluzole blocks the downstream effects of NMDAR; when administered to AD model mice, it significantly enhanced cognition and reduced Aβ_42_, Aβ_40_, Aβ oligomers’ levels and Aβ plaque load [99]. Other drugs that enhance glutamatergic neurons are N-acetylcysteine, gabapentin, lamotrigine, and topiramate.(c)*Reduced GABA function occurs in AD.* γ-aminobutyric acid (GABA) is the primary inhibitory neurotransmitter in the brain, regulating cognition, memory, adult neurogenesis, and circadian rhythm (see above). GABA synthesis occurs via the α-decarboxylation of L-glutamate by glutamic acid decarboxylase (GAD). Reports of studies using GABAergic drugs have given inconsistent results; that may be, in part, because GABA receptors have subunits with different abilities to bind GABA. The dominant finding in AD is a reduced level of GABA in the cerebral cortex and hippocampus. A meta-analysis of 48 reports involving 603 AD patients found a global reduction of GABAergic components in both brain and cerebrospinal fluid (CSF) [100]. Advanced AD with Braak stages V and VI showed the most marked decreases in GABA as compared to earlier stages [101]. In the frontal cortex of AD, there was a reduced level of the long isoform of the Munc18-1 protein, which is important for presynaptic GABA function [102]. Membranes of pyramidal cells in contact with amyloid plaques lacked GABAergic perisomatic synapses, causing decreased neuronal inhibition; their loss may lead to the hyperactivity of such neurons [103]. Thus, treatment to raise GABA levels in AD may contribute to reversing dementia. This may be achieved with vigabatrin, which is an FDA-approved inhibitor of GABA transaminase. The administration of 50 mgs of vigabatrin produced a 40% increase in GABA in human brain [104]. Nicotinamide also raises GABA levels [105].(d)*Reduced adrenergic neurons* are also a factor in AD, in which there was a 50% loss of cells in the rostral locus coeruleus (LC) and a 31% reduction in noradrenaline concentration in the midtemporal cortex [106]. Of the adrenergic neurons in the LC, 49% project to the medial prefrontal cortex (PFC), 28% to the orbitofrontal cortex, and 18% to the anterior cingulate cortex (ACC) [107]. Prazosin and erythropoietin increased the levels of α-2 adrenergic activity [108,109]. Prazosin, an α-1 adrenoreceptor antagonist, prevents the closure of gap junctions and allows communication between neurons, astrocytes, and blood vessels [110]. Erythropoietin regulates α-2 adrenergic activity via cells of the LC, which express erythropoietin receptors [111] and are a major source for control of norepinephrine formation [112].(e)*Reduced serotonergic neurons* are seen in AD, where serotonin (5-HT) levels were significantly lowered in the hippocampal cortex, hippocampus, caudate nucleus, and putamen, and the concentrations of 5-HIAA, a metabolite of 5-HT, were reduced in three cortical areas, thalamus and putamen [113]. Serotonin reuptake inhibitors (SSRIs), e.g., fluoxetine and venlafaxine, increase the availability of serotonin for its binding by the serotonin receptor.

*Drugs useful for reversing the reduced cholinergic, glutamatergic, gabaergic, adrenergic, and serotonergic neurons in AD* include donepezil, galantamine, rivastigmine, pioglitazone, memantine, riluzole, gabapentin, N-acetylcysteine, lamotrigine, topiramate, vigabatrin, prazosin, erythropoietin, fluoxetine, and venlafaxine.

## 4. Combinations That Include Drugs Providing Benefits to More than One Component of Pathogenesis

The cure of AD requires three reasonable assumptions. Firstly, therapy must reverse all or most of the major components of pathogenesis. Others have already suggested this: “We propose that AD could be tackled not only using combination therapies including Aβ and tau, but also considering insulin and cholesterol metabolism, vascular function, synaptic plasticity, epigenetics, neurovascular junction and blood–brain barrier targets” [114]. In this regard, if one component causing, e.g., amyloid, is essential but by itself insufficient, then other subcellular causes must interact with it and, together, produce AD; thus, blocking those other causes might reverse dementia. Secondly, a course of therapy may require a minimum duration. Thirdly, patients would not tolerate treatments requiring more than three drugs. From the list of 29 drugs that are suggested to reverse the 18 elements of pathogenesis, one may suggest ten triple-drug combinations that could be administered sequentially, each combination for either 13 weeks, for a total duration of 2.5 years, or for 26 weeks, for a total duration of five years. It is important to ascertain that the suggested drugs may penetrate the brain.

### Brain Penetration of the Mentioned Drugs

Suggestions for therapies that aim to cure AD are worthless if the drugs in the forms that are available commercially have poor penetration into the brain. Many compounds that are effective, when applied to cultured neurons, do not cross the blood–brain barrier (BBB). However, by using nano-technology to package them in a sufficiently small size, 1–100 nm, they can now cross the BBB into the brain [115]. Those nano-particles may be polymer- or lipid-based and, especially if coated with surfactants, may become effective in the brain. Unfortunately, if unavailable except in a research laboratory, those modifications have no current use, even though they may become available in the future. 

Since most of the drugs suggested above have been commercially available for several decades, many of the studies that document the degree of their penetration into the brain may be from as long ago as 1990—however, that does not diminish their validity. Further, the ideal studies, which are those performed in human brains obtained at autopsy, are rare, although a few more recent studies might document effects on cerebral blood flow in humans using imaging methods. However, the majority of the information about the penetration of the brain would, realistically, have been obtained from studies on animals, which raises the question as to their relevance to the human situation. Important differences between rodent and human brains are described in refs. [116,117].

The following description shows the degree to which each of the drugs penetrates the brain, which must dictate the choice of which triple combinations to use in a clinical trial.

*Atorvastatin and the brain.* Trastuzumab administered to mice induced cognitive defect, as shown by performance in a passive avoidance test [118]. The ^18^F-fluorodeoxyglucose scan showed decreased cerebral glucose metabolism in the frontal lobe of the mice that were rescued by subsequent atorvastatin administration.

*Diclofenac and the brain.* Children undergoing surgery for either inguinal hernia repair or orchidopexy under spinal anesthesia had blood and spinal fluid analyzed after diclofenac was used for pain control [119]. After being injected intravenously, diclofenac was detected as early as 5 min later in CSF at significant concentrations, which were sustained for up to four hours. Since bacterial and virus infections elevate the levels of cytokines in serum and cerebrospinal fluids, Fukada et al. asked whether such high levels of cytokines might alter the integrity of the blood–brain barrier (BBB) and/or blood–cerebrospinal fluid barrier (BCSFB) and subsequently affect brain penetration of antipyretic drugs [120]. They observed that a Shiga-like toxin administered to mice increased the brain:plasma ratio of diclofenac. 

*Doxycycline and the brain.* 6-hydroxydopamine induces a slow loss of dopamine neurons in the substantia nigra. That loss was mitigated by doxycycline [121].

*Erythropoietin and the brain.* Transgenic mice that express human erythropoietin in their brain were subjected to occlusion of the middle cerebral artery; their infarct volume, in comparison to that in wild-type control mice, was decreased by 84%, and the neuronal death was decreased by 54.5% [122]. A similar occlusion in rats, followed by the administration of erythropoietin, also produced a dramatic reduction in the infarct volume and neuronal apoptosis [123].

*Fluoxetine and the brain.* By using cultures of rat cortical neurons, it was shown that fluoxetine inhibited NMDA-evoked currents [124], which would largely be a disadvantageous event. In another disadvantage of this drug, it was found that fluoxetine increased the permeability of the BBB and of proinflammatory microglia [125].

*Folate and the brain.* By using folate receptor antibodies, it was shown in a rat model, that folate uptake is via the choroid plexus and cerebral microvessels [126]. The three isoforms of folate are folic acid, DL-folinic acid, and levofolinate; the same study showed that the highest concentration in the cerebrum and cerebellum was seen with levofolinate.

*Gabapentin and the brain.* Gabapentin is used to control epilepsy, in which condition it promptly elevates brain GABA levels [127], which is advantageous because GABA itself does not cross the BBB [128].

*Galantamine, donepezil, and the brain.* Brain levels of acetyl cholinesterase inhibition for donepezil and galantamine in rat, mouse, and rabbit brains were determined after their subcutaneous application [129]. The data showed that for a similar degree of brain acetyl cholinesterase inhibition, 3–15 times higher galantamine than donepezil doses are needed. A randomized, placebo-controlled study in humans with AD showed that patients treated with galantamine had significantly better cognitive functions than those treated with placebo [130].

*Gemfibrozil and the brain.* Gemfibrozil was shown to enter the brain in a mouse model of Parkinson’s disease, in which the oral administration of the human-equivalent dose protected dopaminergic neurons in the substantia nigra [131].

*Insulin and the brain.* Patients with MCI who progressed to dementia had a reduced cerebral metabolic rate of glucose metabolism [38]; that is because cerebral resistance to insulin is heightened in diabetics [39]. Another action of insulin is a reduction in the proinflammatory M1 microglial phenotype [40].

*Lecanemab and the brain.* Certainly, lecanemab enters the brain; otherwise, it would have no clinical efficacy. However, the degree by which lecanemab enters the brain is uncertain because the presence of edema and hemorrhage, as shown by MRI imaging in a clinical study, was no different from that seen with placebo, nor were there measurable effects on CSF biomarkers [132]. 

*Lithium and the brain.* Lithium produces a reversal of the insulin resistance that is seen in AD [133], and it restored insulin sensitivity in diabetic rats [134]. Additionally, it increased the transport of glucose induced by insulin by ~2.5-fold [135].

*Melatonin and the brain.* Melatonin is an endogenous brain product, and there are melatonin receptors expressed widely in the brain that are relevant to circadian physiology [136].

*Memantine and the brain.* Memantine infused into rats achieved a brain concentration that was 44-fold higher than in the serum [137].

*N-acetyl cysteine (NAC) and the brain.* NAC produced a 49.7% reduction in the infarct volume that was caused by middle cerebral artery occlusion in rats [138].

*Nicotinamide and the brain.* Nicotinamide was used in two dosages, 50 or 500 mg/kg, in rats subjected to percussion injury of their brains; only the high dosage produced benefit for tissue sparing, and neither dosage affected glial fibrillary acid protein or ubiquitin hydrolase levels [139].

*Pioglitazone and the brain.* In rats subjected to traumatic brain injury, pioglitazone produced a decrease of 55% in the brain lesion size and complete protection of cognitive function assessed using the Morris Water Maze test [140].

*Prazosin and the brain.* In prazosin-treated and saline-treated rats with experimental autoimmune encephalitis, increased brain edema was associated with increased vesicular content of capillary endothelium. In prazosin-treated rats with no clinical signs, the vesicular content of endothelial cells was comparable to that found in control animals [141].

*Prednisone and the brain.* Prednisone is widely used to treat cerebral edema, so it must both rapidly and extensively enter the brain.

*Pyridoxine and the brain.* Radioactively labeled pyridoxine was shown to readily enter the CSF, choroid plexus, and brain [142].

*Rapamycin and the brain.* Following lipopolysacharide injection into rats in order to induce encephalopathy, rapamycin was found to inhibit neuroinflammation, normalize brain vascularity and the BBB, and several brain metabolites [143].

*Rasagiline and the brain.* Neuroprotective activity of rasagiline in the brain is associated with protection of mitochondrial viability and the mitochondrial permeability transition pore; it does so by activating Bcl-2 and protein kinase C (PKC), and down regulating pro-apoptotic FAS and Bax [144].

*Riluzole and the brain.* Riluzole was administered to rats with traumatic brain injury, and 48 h after the trauma, brains were removed; in rats with trauma at only 1 mm penetration, hemispheric swelling, cerebral water content of the traumatized hemisphere, and cortical contusion volume were significantly reduced by riluzole compared with controls (*p* < 0.05) [145].

*Rivastigmine and the brain.* Clinical investigations of rivastigmine administered at doses of 6–12 mg/day significantly improved cognition, as measured by the ADAS-Cog score, and activities of daily living, as measured by the Progressive Deterioration Scale [146].

*Venlafaxine and the brain.* In clinical studies of patients with depression, venlafaxine was found to penetrate the CSF with high ratios of CSF:plasma [147].

*Vigabatrin and the brain.* Cerebral levels of vigabatrine were obtained by microdialysis catheters placed in patients with traumatic brain injuries; drug levels were highest in samples taken the closest to the lesions [148].

*Vitamin C and the brain.* Vitamin C is considered to be a vital antioxidant molecule in the brain, where it helps maintain integrity and function of several processes, including neuronal maturation and differentiation, myelin formation, synthesis of catecholamine, modulation of neurotransmission, and antioxidant protection [149].

## 5. Construction of a Clinical Trial to Demonstrate Safety and Efficacy

The two durations (2.5 or 5.0 years) should undergo concurrent trials because it is unknown whether an adequate duration would be obtained after exposures of three or six months. Although six months of treatment is probably adequate, the possibility that the shorter duration of 13 weeks suffices is supported by studies in ovariectomized rats showing success from brief durations of treatment. Choline acetyltransferase activity in the frontal cortex (+20.1%) and olfactory bulbs (+30.4%) increased after only two weeks of treatment with estrogen plus progesterone [150]; only 40 days of treatment with estradiol enhanced working memory [151]; and 24 h after receiving estradiol, there was already significant increase of dendritic spines in pyramidal cells of cortical layer 111 [152]. Unfortunately, no human study has assessed what minimum duration of treatment produces similar benefits.

Clinical trials would provide for the termination of treatment if earlier success were declared by a data safety monitoring board (DSMB). A neuropsychologist would administer cognition tests at baseline, then annually and at discontinuation of therapy. The trial would randomize patients to receive either the suggested therapies sequentially or, for equipoise, triple therapy with donepezil, memantine, and lecanemab. Calculation of the required n would assume a 10% annual loss of participants caused by the discontinuation of treatment or death and an intention-to-treat analysis, showing 50% more cure from the experimental treatments as compared to the equipoise treatment. Effect size and power will be calculated by a biostatistician based on a ≥30% rate of cure as compared with that achieved by the equipoise arm.

Inclusionary criteria would be age 60–75 years and a clinical diagnosis of spontaneous AD based upon the assessed cognitive loss plus a brain scan showing amyloid deposition. Exclusionary criteria would include untreated BP ≥ 140/85, known cerebrovascular disease, diabetes, active systemic cancer, and untreated hepatitis C. Active treatment would comprise triple combinations, each administered for 13 or 26 weeks in any order. Counted as one drug would be folate plus pyridoxine and pioglitazone plus either fluoxetine or lithium; statins would be either atorvastatin or simvastatin.

In order to monitor for adverse events in this elderly population, participants will be seen every 4 weeks for direct questioning about side effects, physical examination, and laboratory studies. If intolerance or unacceptable side effects occur from one triple-drug combination, that would be stopped, and the next combination tried; if that is also not tolerated, the patient will withdraw from the study. At the 4-weekly visit, there will be direct questions asked concerning adherence. At those visits, patients will show the bottles of medications used in the trial and the residual pills will be counted, which will provide an accurate measure of adherence. At a time point between those visits, the Research RN will telephone every subject to enquire about problems and encourage continued participation.

There are 3654 possible combinations of three drugs taken from a list of 29 different drugs. The following are ten suggested triple combinations, but of course, hundreds of others are possible:

1, rapamycin, intranasal insulin, fluoxetine; 2, memantine, riluzole, venlafaxine; 3, doxycycline, simvastatin or atorvastatin, prednisone; 4, lithium, diclofenac, melatonin; 5, light therapy, galantamine, N-acetyl cysteine; 6, lecanemab, donepezil, vigabatrin; 7, gabapentin, prazosin, erythropoietin by weekly injection; 8, rasagaline, caffeine, gemfibrozil; 9, folate/pyridoxine, vitamin E, vitamin C; 10, nicotinamide, pioglitazone plus either fluoxetine or lithium. By using the online Drug Interaction Checker published by Brigham and Women’s Hospital, the components of each suggested triple-drug combination, including the equipoise combination, were examined for potential drug–drug interactions, and none was found. For each participant in the clinical trials, the drug combinations provided must also be checked for interactions with whatever other medications that are being used for comorbidities. However, drug–drug interactions are uncommon: a Swedish study of 8214 patients who had received ≥2 prescriptions over a 14-month period showed that only 167 (2.0%) had an interaction with potentially serious consequences [153].

The lowest dosages, which will be used for each of the above drugs, are shown in the following (alphabetic) list: aducanumab (1 mg/kg increasing to 10 mg/kg IV q4wk), atorvastatin (10 mg qd), caffeine (1 glass tea qd), diclofenac (75 mg delayed release tab. qd), doxycycline (100 mg qd), erythropoietin (50 units/kg IM, q7 days), fluoxetine (10 mg qd), folate (5 mg qd), galantamine (4 mg bid), gabapentin (300 mg tid), gemfibrozil (300 mg qd), intranasal insulin (20 iu, qd), lecanemab (10 mg/kg IV q2w), lithium (150 mg qd), melatonin (5 mg qHS), memantine (5 mg qd), N-acetyl cysteine (effervescent tab: 600 mg once daily); nicotinamide (100 mg qd), pioglitazone (15 mg qd), prazosin (1.0 mg qd), prednisone (2.5 mg qd), pyridoxine (25 mg qd), rapamycin (0.5 mg qd), rasagaline (0.5 mg qd), riluzole (50 mg qd), rivastigmine (1.5 mg bid), venlafaxine (37.5 mg qd), vigabatrin 25 mg bid), vitamin C (1000 mg qd).Many of these drugs have potentially serious side effects, so clinical trials must have an assigned clinical pharmacologist whose function will be to assess whether or not an individual patient should receive a given triple-drug combination. Removing that triple combination will lower the total exposure of many participants to all of the drugs; that could be beneficial if a substudy of such participants showed that even lower exposure to the triple-drug combinations could produce a higher cure rate compared with equipoise treatment.Treatment for conditions that occurred during one year, either before or after the appearance of AD. Finally, it is the role of risk factors whose first presence or worsening was ±1 year when AD first appeared. Those elements were examined in three analyses [154,155,156]. Twelve risk factors for AD included inadequate exercise, hearing loss, and infrequent social contact [154], plus six others, for which Barnes and Yaffe gave the following population prevalences: physical inactivity 32.5%, smoking 20.6%, depression 19.2%, midlife hypertension 14.3%, mid-life obesity 13.1%, and diabetes 8.7% [155]. Twelve reports involving 6865 participants showed metabolic syndrome as another risk factor [156], and the Italian Longitudinal Study on Aging found that it approximately doubled the risk of dementia [157]. Another report included metabolic syndrome and added nutrient deficiencies, traumatic head injuries, and occupational exposure to toxins [158].

In brief, when one or more of the conditions indicated above occur concomitantly with depression, then their correction *plus* the other approaches suggested in this article might assist in curing dementia. Standard of care for the particular comorbidity should be administered.

## 6. Summary

Comprehensive treatment aimed at curing Alzheimer’s dementia may require addressing all of the components of its pathogenesis.

This article lists 18 components that have an important pathogenetic role, and for each component, it also provides beneficial treatments. Those treatments aggregate 29 drugs, so they cannot be administered concurrently. Instead, they can be provided as triple-drug combinations administered sequentially, with each combination administered for either 13 weeks, so that ten triple combinations would require a duration of 2.5 years, or for 26 weeks and a total duration of 5 years.

A clinical trial is necessary in order to validate the safety and efficacy of the ten triple-drug combinations.

## Figures and Tables

**Table 1 ijms-25-03909-t001:** List of causal factors, some of which may participate in pathogenesis of dementia in an individual patient with Alzheimer’s disease. Curative approach #1 addresses all factors.

Conditions to Address
Deposition of Aβ/amyloid
Deposition of tau protein
↑reactive oxygen species
↓Wnt/β-catenin
↓TGFβ
↓EMT
Abnormal cerebral microvascularity
Insulin resistance (cerebral)
↑intracellular Ca^2+^
↓Autophagy
↓UPR
↑inflammation
miRNA
Mitochondrial dysfunction
↓BDNF
Synaptic and neuronal dysfunctions
↑ApoE4ε
Abnormal circadian rhythm

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
