# Peer review of "Personalized, Precision Medicine to Cure Alzheimer’s Dementia: Approach #1"

_ijms, 2024, doi:10.3390/ijms25073909_

Round 1

Reviewer 1 Report

Comments and Suggestions for Authors

In this paper the author describes the approach 1 (indicated in the title) to cure Alzheimer’s dementia. For this reason, 18 factors that cause Alzheimer's disease and the 31 different treatments available to address them have been described. Considering that curing Alzheimer's disease requires basic assumptions (i- therapy must reverse all or most of the major factors of pathogenesis; ii- a course of therapy may require a minimum duration; iii- patients would not tolerate treatments requiring more than three drugs), the author proposes triple drug combinations that could be administered for a13 or 26 weeks in any order for a total duration of 2.5 or five years. The author also suggests inclusion/exclusion criteria for AD patients and the dosage to be used for each drug. Finally, the author outlines a randomized clinical trial necessary to evaluate the safety and efficacy of the proposed treatments.

The observations here are very interesting, but there are comments to be made.

Ø  Acronyms should be explained as soon as they are inserted into the text from the abstract (AD, DSMB, TCA….).

Ø  There are misspelled words (line 156, page 7);

Ø  Braak stage III” needs to be explained in the text (line 212 page 9).

Ø  The text is not well organized into paragraphs and sub-paragraphs. The author should better organize the paragraph 3.10 into sub-paragraphs (3.10.1….); the sub-paragaraph 3.29 should be a new paragraph (4.0…)

Ø  There is no need to divide the summary 6.11 into 6.12, 6.13.

Ø  The author should maintain consistent text formatting throughout the manuscript.

Author Response

Responses to Reviewer 1.

  1. Expand abbreviations at their first use in the text.

Response. Agreed! I counted 20 abbreviations already expanded in the original text. Needing expansion, however, are (listed according to their appearance in the text): transforming growth factor, randomized clinical trial, amyloid precursor protein, tricyclic antidepressants, cerebrospinal fluid, prefrontal cortex, anterior cingulate cortex. They are now expanded in the revised text. I have not expanded several enzymes or cytokines for which abbreviations are in common usage (e.g., GSKβ).

  1. There are misspelled words (line 156 p7).

Response. I thank the reviewer for noting this: in fact, several lines of text are missing from the text provided to the Reviewer, creating the impression of misspelling. It is unclear how that happened. The revised text has those missing lines, outlined in red.

  1. Braak stage 3 needs to be explained.

Response. Braak stages in Alzheimer’s disease have been in continuous use for several decades. I disagree that they need explanation.

  1. Re-organize section 3.10.

Response.I have done so according to the Reviewer’s recommendations.

  1. Maintain consistent text formatting.

Response. I have attempted to do so but I am not very good with this task so I request to leave its accomplishment to the Copy Editor.

Reviewer 2 Report

Comments and Suggestions for Authors

The paper outlines an intricate protocol for administering 31 medications in sets of three, staggered over time. While the concept is innovative, the rationale behind the selection of specific drug combinations and their sequencing is not fully elucidated. The author is encouraged to elaborate on the rationale for each medication's selection and the sequence's foundation. Additionally, the presentation of the treatment plan could be clearer, potentially through a visual aid like a timeline or flowchart, to better convey the process of sequential drug administration. The format of the document should also adhere to the journal's prescribed template.

The author notes an absence of drug-drug interactions based on an online tool, yet the administration of 31 drugs, even in groups of three, presents considerable safety and pharmacokinetic concerns. A deeper investigation into potential interactions and their clinical significance is necessary, with attention to clinical pharmacy methods for detecting drug interactions.

The paper should also provide a more detailed strategy for monitoring adverse effects, especially given the elderly demographic and the risks associated with multiple medications.

The document identifies medications aimed at addressing the 18 pathogenic factors of Alzheimer's dementia. Nonetheless, the evidence supporting the effectiveness of each drug for AD treatment is not uniformly presented. The author should compile and present the evidence supporting the inclusion of each medication in the treatment plan.

The author concedes that patients may not be able to manage more than three medications simultaneously, but the prospect of long-term treatment with multiple drugs brings up issues of patient adherence and tolerance. The paper would benefit from a discussion on methods to maintain patient commitment to the treatment plan throughout the proposed duration of 2.5 to 5 years.

The proposed clinical trial is described with specific criteria for inclusion and exclusion, but the paper does not explain the basis for these criteria or consider how they might affect the broader applicability of the trial findings.

The paper should also elaborate on the statistical power and the necessary sample size to ensure the study can reliably detect the anticipated effects.

The author briefly addresses the treatment of comorbid risk factors but does not thoroughly examine how these might interact with the proposed treatment for Alzheimer's dementia. An extensive discussion on managing these comorbidities and their potential influence on treatment outcomes is needed.

While the paper lists the pathogenic factors and the drugs that correspond to them, it does not deeply explore the mechanisms through which these drugs are supposed to alter the disease's progression. A more comprehensive explanation of the pathophysiological basis for including each drug would enhance the paper.

Comments on the Quality of English Language

The English quality of the manuscript is fine

Author Response

  1. Elaborate the rationale for each medication’s selection; and produce a visual aid to illustrate the process.

Response. This is already described in lines 72-333. I cannot see how a visual aid will substitute for a careful reading of those 261 lines of text! The rationale is further discussed in the text lines 364-452. Surely, enough is enough?

  1. The administration of 31 drugs, even in groups of 3, presents safety and pharmacokinetic concerns. There needs to be a deeper investigation into potential interactions with attention to clinical pharmacy methods for detecting them.

Responses. First, the Reviewer notes that an “online tool was used to document absence of drug-drug interactions”. I want to point out that the online tool that I used is published by the Brigham and Women’s hospital in Boston. That is generally acknowledged as one of the top hospitals in the USA, and it is a major teaching hospital of the Harvard medical school. I believe, therefore, that it might be accepted as authoritative, that no drug-drug interactions were found among the possible combinations mentioned in the article. Nevertheless, the reviewer suggests that there should be more attention to clinical pharmacy methods for detecting drug interactions. In that regard, I want to point out that serious drug-drug interactions are uncommon. A Swedish study of 8214 patients who had received ≥2 prescriptions over a 14 months’ period, showed that only 167 (2.0%) had an interaction with potentially serious consequences (Astrand B, et al., Eur J Clin Pharmacol. 2006;62:749). In a drug claims database that included approximately 2.9 million patients with more than 30 million prescriptions dispensed in the 12-month period from September 2001 through August 2002, there were only 65,544 drug pairs with clinically relevant drug-drug interactions (0.2% of total claims) and clinical pharmacist review reduced this to only 0.04% of total prescription claims (Peng C, et al. J Managed Care-Specialty Pharmacy. 2003;9:532).

  1. Provide a more detailed strategy for monitoring adverse events, particularly in the elderly patients.

Response. The following will be added to the text: ‘Subjects will be seen every 4 weeks when direct questioning, physical examination, and laboratory studies, will monitor for adverse events.’

  1. Provide the evidence supporting the use of each medication.

Response.. This already exists, specifically, for each drug (see lines 364-452); and further evidence is also provided in the text dealing with each of the 18 causal elements

  1. Provide a plan to support patient commitment to maintaining the treatment program.

Response. This is a very important criticism and is greatly appreciated. The following will be added to the text describing the clinical trial:

’Also every 4 weeks there will be direct questions asked concerning adherence. At those visits, patients will show the bottles of medications used in the trial and the residual pills will be counted, which will provide an accurate measure of adherence. At a time point between those visits, the Research RN will telephone every subject to enquire about problems, and encourage continued participation.’

  1. Explain the basis for the Inclusionary and Exclusionary criteria in the clinical trial.

Response. Those criteria are standard ones, used in most protocols for clinical trials. Readers familiar with protocols for clinical trials, will not see a problem; and I see no purpose in offering a detailed defense for each criterion.

  1. Elaborate on calculating the n and power of the clinical trial.

Response. As indicated in lines 469-470, these will be calculated by a Research Biostatistician associated with the trials.

  1. Discuss management of comorbidities and their influence on treatment outcomes.

Responses. Several comorbidities are risk factors for dementia, e.g., diabetes, hypertension, hyperlipidemias, etc. If present in an individual participant, they will be addressed by standard therapies with which the trial’s Principal Investigator will surely be familiar, so I see no need for elaboration. There is a problem regarding treatments for depression because several reports suggest that antidepressants might enhance the risk for subsequent dementia; however, if present in subjects with established dementia, depression must be addressed with standard treatments.

  1. Expand the explanation of the pathophysiological basis supporting the use of each drug.

Response: In fact, this comment and its response are the same as for comment #1.

Round 2

Reviewer 2 Report

Comments and Suggestions for Authors

I consider the manuscript proper to be accepted. The authors clarified all my inquiries